# In Situ Pinpoint Photopolymerization of Phos-Tag Polyacrylamide Gel in Poly(dimethylsiloxane)/Glass Microchip for Specific Entrapment, Derivatization, and Separation of Phosphorylated Compounds

**DOI:** 10.3390/gels7040268

**Published:** 2021-12-16

**Authors:** Sachio Yamamoto, Shoko Yano, Mitsuhiro Kinoshita, Shigeo Suzuki

**Affiliations:** 1Faculty of Pharmacy, Kindai University, 3-4-1 Kowakae, Higashiosaka 577-8502, Osaka, Japan; syoko.yano@kindai.ac.jp (S.Y.); m-kino@phar.kindai.ac.jp (M.K.); suzuki@phar.kindai.ac.jp (S.S.); 2Antiaging Center, Kindai University, 3-4-1 Kowakae, Higashiosaka 577-8502, Osaka, Japan

**Keywords:** online preconcentration, online derivatization, in situ photopolymerization, Phos-tag acrylamide

## Abstract

An improved method for the online preconcentration, derivatization, and separation of phosphorylated compounds was developed based on the affinity of a Phos-tag acrylamide gel formed at the intersection of a polydimethylsiloxane/glass multichannel microfluidic chip toward these compounds. The acrylamide solution comprised Phos-tag acrylamide, acrylamide, and *N,N*-methylene-bis-acrylamide, while 2,2′-azobis[2-methyl-N-(2-hydroxyethyl)propionamide] was used as a photocatalytic initiator. The Phos-tag acrylamide gel was formed around the channel crossing point via irradiation with a 365 nm LED laser. The phosphorylated peptides were specifically concentrated in the Phos-tag acrylamide gel by applying a voltage across the gel plug. After entrapment of the phosphorylated compounds in the Phos-tag acrylamide gel, 5-(4,6-dichlorotriazinyl)aminofluorescein (DTAF) was introduced to the gel for online derivatization of the concentrated phosphorylated compounds. The online derivatized DTAF-labeled phosphorylated compounds were eluted by delivering a complex of phosphate ions and ethylenediamine tetraacetic acid as the separation buffer. This method enabled sensitive analysis of the phosphorylated peptides.

## 1. Introduction

Post-translational modifications (PTMs), which involve covalently attaching chemical entities to the side chains of modifiable residues, act as molecular switches that allow cells to respond to diverse conditions [1,2]. PTMs play a vital role in the control of protein activity, stability, and subcellular localization, thereby contributing to intracellular regulation [3,4]. Among these modifications, protein phosphorylation is one of the most studied PTMs. More than 30% of eukaryotic proteins were observed to be phosphorylated [5]. The addition of a phosphate group to the amino acid residue disrupts the existing electrostatic interactions and creates new hydrogen bonds in the substrate protein. As a consequence, the protein structure is significantly altered, which affects the stability, activity, and subcellular localization of the protein and its ability to specifically regulate a cellular process [6,7,8,9]. Mass spectrometry (MS) is the most used method for analyzing phosphorylation of protein and can profile thousands of proteins in a single analysis [10]. The presence of non-phosphorylated peptides in sample solution suppressed the ionization of phosphorylated peptides, thereby preventing the detection of phosphorylated peptides through MS [11]. Typically, specificity and sensitivity are important issues encountered during the detection of a phosphorylated compound.

Derivatization represents sample preparation with the primary aim of increasing the sensitivity. Phosphorylated peptides are commonly derivatized with fluorescent reagents to enable their detection with laser-induced fluorescence detectors, the sensitivity of which is approximately two to three orders of magnitude higher than that of the most frequently used UV-spectrophotometric absorbance detectors, which do not require derivatization and are capable of selective analysis of variable compounds in some cases [12]. Another challenge in phosphoproteome is to develop a specific concentration method to entrap low-abundance phosphorylated peptides [13]. Specific phosphorylated peptide concentration methods, such as phosphoramidite chemistry [14], immobilized metal ion affinity chromatography [15], strong cation exchange chromatography [16], metal oxide chromatography [17], and aliphatic hydroxy acid-modified metal oxide chromatography [18,19] were utilized phosphoproteome. These methods have to be utilized for the analysis of phosphopeptides by capillary electrophoresis (CE) where the sensitivity of the used detectors and separation of the employed CE are insufficient for the direct analysis of peptides present at low concentration levels in complex mixtures. Microchip electrophoresis (ME) has utilized for the short-time analysis of proteins and peptides [20] due to minimal sample consumption. Several efficient enrichment methods have been developed to ME analysis, including isotachophoresis [21], multiple preconcentration techniques [22,23], and hydrogel concentration methods [24,25,26].

We have previously reported a specific phosphate affinity ligand (1,3-bis[bis(pyridin-2-ylmethyl)amino]propan-2-ol) referred to as Phos-tag [27,28,29] acrylamide gel, which is formed at the channel-cross utilizing in situ photopolymerization. The fluorescein isothiocyanate (FITC)-labeled phosphopeptides in a mixture of peptides were specifically concentrated in the Phos-tag acrylamide gel by applying a voltage across the Phos-tag gel. This method delivered sensitive analysis of the phosphorylated peptides in a complex peptide mixture [30]. However, due to the employment of a single cross-pattern microchip in the previous study, fluorescent labeling of the samples was necessary as pretreatment for sensitive detection.

Here, we describe an improved method for the online preconcentration, derivatization, and separation of phosphorylated compounds based on the affinity of the Phos-tag acrylamide gel formed at the intersection of a polydimethylsiloxane (PDMS)/glass multichannel microfluidic chip toward these compounds. Our PDMS/glass microchip has three channel crossing points and eight reservoirs. We placed five types of solutions such as samples and fluorescent reagents in these eight reservoirs, and achieved concentration, fluorescent labeling, separation, and detection by switching the voltage. The Phos-tag acrylamide gel was formed around one of the three channel crossing point by irradiation with a 365 nm LED laser. By fabricating Phos-tag acrylamide gel around the channel crossing point, the volume of the phosphorylated peptide that can be captured was increased by 10 times, and utilized as a wide reaction field for online fluorescent derivatization. The unlabeled phosphorylated peptides were specifically concentrated in the Phos-tag acrylamide gel by applying a voltage. After the entrapment of the unlabeled phosphorylated compounds in the Phos-tag acrylamide gel, 5-(4,6-dichlorotriazinyl)aminofluorescein (DTAF) was introduced to the gel for online derivatization of the concentrated phosphorylated compounds. The online derivatized DTAF-labeled phosphorylated compounds were eluted by delivering a complex of phosphate ions and ethylenediamine tetraacetic acid (EDTA) as the separation buffer. Sensitive analysis of the phosphorylated peptides was achieved by this method.

## 2. Results and Discussion

A method was developed for the capture, concentration, and online derivatization of phosphopeptides on a Phos-tag polyacrylamide gel followed by separation and detection on a PDMS/glass microfluidic chip. Figure 1 shows the PDMS/glass microchip used in this experiment and the Phos-tag acrylamide gel in the channel crossing point. This method needed to achieve the following four steps: (1) fabrication of Phos-tag gel around the channel crossing point; (2) concentration of the phosphopeptides; (3) online derivatization of the phosphorylated compounds in the Phos-tag gel using a fluorescent reagent; and (4) elution and separation of the phosphorylated compounds. In order to achieve these four steps, it is necessary to introduce various solutions into the Phos-tag gel. Also, in step (3), excess fluorescent reagent must be removed from the channel and gel after online derivatization for high sensitivity detection. Therefore, we fabricated a PDMS microchip with three channel cross points and eight reservoirs, and investigated above four steps to achieve only by applying voltage.

### 2.1. Fabrication of the Affinity Matrix

To ensure the efficient trapping of the phosphorylated compounds by the gel, Phos-tag should be effectively incorporated in the acrylamide gel and maintained the activity in the gel. The concentration and molar ratio of acrylamide/bis-acrylamide determine the pore size and mechanical strength of the gel. In our previous study, 0.04% Phos-tag acrylamide in 20%T/20%C was used for the fabrication of the Phos-tag acrylamide gels in polymethylmethacrylate (PMMA) chips. T% and C% represent the percentage of acrylamide compounds in gel solution and the percentage of *N,N′*-methylene-bis-acrylamide in acrylamide compounds, respectively. First, we prepared the acrylamide gels according to our previous report. However, the gels fabricated in the PDMS/glass microchip broke when a voltage of 1 kV was applied. We prepared various concentrations of acrylamide gels in the range of 5–30% and their stability was determined by applying a voltage of 1 kV across the gels. Sufficient strength was obtained at a concentration exceeding 30%T and 20%C, and no breakage was observed. The ability of the Phos-tag group to trap phosphorylated compounds increased by using high concentrations of Phos-tag acrylamide in the gel. However, higher concentrations (>0.05%) of Phos-tag produces a white turbidity gel. Therefore, 0.04% Phos-tag acrylamide containing 30%T/20%C was used for fabricating the Phos-tag polyacrylamide gels. The gels were fabricated within 5 min by irradiation with LED. After fabrication, the 10 mM MnCl_2_ were introduced to Phos-tag acrylamide for 3 min followed by another channel wash with 25 mM sodium borate (pH 8.5) for 3 min.

### 2.2. Entrapment and Concentration of Phosphorylated Compounds

To investigate the utility of this method, we chose β-casein monophosphopeptide as a model peptide and FITC, DTAF, and 4-fluoro-7-nitrobenzofurazan (NBD-F) as acidic reagents, which have appreciable reactivity toward amines for online derivatization. The Phos-tag polyacrylamide gel was fabricated at the channel-cross, C3 (Figure 1c) by photopolymerization. The Phos-tag groups in the gel had their activity by introducing 10 mM MnCl_2_ and all channels were washed by using 25 mM sodium borate buffer (pH 8.5). The reservoirs R1, R5, and R8 were filled with a 100 mM Tris phosphate buffer (pH 7.0)/100 mM EDTA solution, 200 mM sodium borate buffer (pH 11.0), fluorescent reagent, and β-casein phosphorylated peptides, respectively. The other reservoirs (R2, R3, R4, R6, and R7) were filled with a 25 mM solution of sodium borate (pH 8.5).

To accurately calculate the concentration efficiency, an online concentration experiment was performed using the β-casein phosphorylated peptides pre-labeled with FITC, DTAF, and NBD-F. The FITC, DTAF, and NBD-F-labeled β-casein phosphorylated peptides were introduced to the Phos-tag gel by applying 150 V between R8 and R4. Figure 2 shows the time-courses of the specific concentrations of the three fluorescent reagents-labeled β-casein phosphorylated peptides in the Phos-tag polyacrylamide gel at the channel crossing point, C3. After washing and coordination of the Phos-tag polyacrylamide gel with Mn^2+^ ions, no fluorescence was detected in the gel. On applying a voltage, the fluorescence intensity of the Phos-tag polyacrylamide gel gradually increased in the presence of the FITC, DTAF, and NBD-F-labeled β-casein phosphorylated peptides. We also monitored the fluorescence intensity of 10^−8^ and 10^−9^ M solutions of FITC, DTAF, and NBD-F-labeled β-casein phosphorylated peptides under identical conditions without fabricating the Phos-tag polyacrylamide gel as reference, and the fluorescence intensities reached a maximum state as depicted by the straight lines in Figure 2. The 10^−9^ M DTAF-labeled β-casein started fluorescing at 0.4 min and the fluorescence intensity increased up to 3 min and then reached a plateau. The 10^−11^ M DTAF-labeled β-casein started fluorescing at 0.2 min and the fluorescence intensity increased gradually until 2.7 min before reaching a plateau. The fluorescence intensity of 10^−11^ M β-casein in the Phos-tag polyacrylamide gel was nearly identical to that of 10^−9^ M β-casein in the absence of the Phos-tag polyacrylamide gel. This implies that the concentration factor obtained using this method is approximately 100-fold at the nanomolar level. FITC-labeled β-casein (FITC concentration: 10^−9^ M) started fluorescing at 0.3 min and the fluorescence intensity reached a maximum at 0.8 min, followed by a decrease and then reaching a plateau after 2 min. At an FITC concentration of 10^−10^ M, the FITC-labeled β-casein started fluorescing at 0.3 min and the fluorescence intensity plateaued at 1 min. The fluorescence intensity of 10^−9^ M β-casein in the Phos-tag polyacrylamide gel was nearly identical to that of 10^−8^ M β-casein in the absence of the Phos-tag polyacrylamide gel. Additionally, the fluorescence intensity of 10^−10^ M β-casein in the Phos-tag polyacrylamide gel was approximately 2 times higher than that of 10^−9^ M β-casein in the absence of the Phos-tag polyacrylamide gel. This implies that the concentration factor obtained using this method is approximately 10-fold at the nanomolar level. At NBD concentrations of 10^−9^ and 10^−10^ M, NBD-F-labeled β-casein started fluorescing after applying a voltage and the fluorescence intensity increased up to 2 min before reaching a plateau. The fluorescence intensity of 10^−9^ and 10^−10^ M β-casein in the Phostag polyacrylamide gel was nearly identical to that of 10^−8^ M β-casein in the absence of the Phos-tag polyacrylamide gel. The time-coursed changes in the fluorescence intensity of NBD-F-labeled β-casein in the Phos-tag gel were not concentration dependent, and there was almost no improvement in the sensitivity. It is plausible that NBD-F was adsorbed on the Phos-tag gel and inhibited the capture of the phosphate compounds by Phos-tag. Therefore, FITC and DTAF were selected as the reagents for online derivatization. In Figure 2, the fluorescence intensity was measured using three fluorescent reagents with the same concentration, but the result was that only FITC had twice the fluorescence intensity. Since we attached a filter for detecting FITC to the microscope, it is considered that the fluorescence intensity of DTAF and NBD-F—which deviate from the excitation wavelength and fluorescence wavelength of FITC—was relatively low.

### 2.3. Online Fluorescence Derivatization

We investigated the feasibility of online fluorescent labeling by monitoring the fluorescence intensity at the channel crossing point. Figure 3 shows the time-course of the changes in the fluorescence intensity of Phos-tag acrylamide gel utilizing 10^−7^ M DTAF or FITC as the online fluorescent reagents. We first introduced 10^−9^ M monophosphorylated β-casein by applying a voltage (150 V) between R8 and R4 for 3 min. Subsequently, the voltage was switched to 100 V and applied between R7 and R4 for 5 min to introduce the DTAF or FITC and the fluorescence intensity was monitored. After 5 min, the voltage was switched to 150 V between R2 and R6 (R6 functioning as the anode) for washing out the excess fluorescent reagents from the Phos-tag acrylamide gel and channel. Fluorescence was detected at 3.0 min at the channel crossing point in the presence of DTAF and the fluorescence intensity increased linearly until 3.8 min before reaching a plateau. The step involving the cleaning of the channel and Phos-tag acrylamide gel extended for 5 min, during which the DTAF in the C2-C3 channel reached the gel, and the fluorescence intensity increased temporarily until 6.9 min. After 6.9 min, the washing buffer (25 mM sodium borate, pH 8.5) reached the Phos-tag acrylamide gel and the fluorescence intensity decreased and reached a plateau after 7.7 min. Thereafter, the fluorescence remained constant. When only 10^−6^ M DTAF was continuously introduced into the gel, the fluorescence intensity increased. However, the fluorescence intensity decreased to the baseline after 4 min (Appendix A). This result indicated that the fluorescence intensity maintained a constant value after 7.7 min during the online derivatization. On the other hand, the fluorescence at the channel crossing point in the presence of 10^−7^ M FITC was detected at 1.1 min and the fluorescence intensity increased almost linearly until the washing step. After 5 min, the fluorescence intensity temporarily increased until 7.0 min and then decreased linearly. Finally, the fluorescence intensity decreased to the baseline after 13.8 min. In order to show this decrease in fluorescence intensity in detail, Appendix A shows the time course of fluorescence intensity from 1 min after the start of measurement and 1 min before the end of measurement. There was a large difference in the fluorescence intensity obtained in Figure 3, which is probably due to the filter set explained in Figure 2 These results indicated that DTAF was useful for online sample derivatization. Therefore, we selected DTAF as the reagent for online sample concentration, derivatization, and separation.

### 2.4. Elution Step

To release the online preconcentrated and DTAF derivatized phosphorylated compounds from the Phos-tag acrylamide gel, it is necessary to dissociate the coordinate bonds between the two divalent metal ions attached to the phosphate groups. Since the sample capture capacity differs depending on the type of metal ion used [31], the most suitable metal for highly efficient concentration and quick desorption was selected. Elution with phosphate ions and EDTA is the most robust method for releasing the phosphorylated compounds trapped in the Phos-tag acrylamide gels [32]. In a previous study, we utilized 100 mM phosphate buffer/10 mM EDTA (pH 2.0) for the elution of FITC-labeled phosphorylated compounds [30]. However, sufficient elution of the DTAF-labeled monophosphorylated β-casein was not achieved. Therefore, in this study, we utilized 100 mM Tris phosphate/100 mM EDTA (pH 7.0) as the dissociation solvent. Figure 4 shows the time-courses of the changes in the fluorescence intensity of prelabeled 10^−9^ M DTAF-monophosphorylated β-casein in the presence of Phos-tag acrylamide gel coordinated with various divalent metal ions. We used Mn^2+^, Zn^2+^, Cu^2+^, and Ni^2+^ ions for coordinating with the Phos-tag acrylamide gel and introduced the elution buffer. When DTAF-labeled monophosphorylated β-casein was introduced to the Phos-tag acrylamides coordinated with the various metal ions, the fluorescence intensity increased with the concentrations of the phosphorylated compounds. After 5 min, we changed the voltage before introducing the elution buffer. The fluorescence intensity decreased in the presence of Phos-tag acrylamide coordinated to metal ions upon introducing the elution buffer. The ability of Phos-tag to capture phosphorylated compounds differs depending on the phosphorylation site and molecular weight [33]. When Cu^2+^ was used for coordination, the fluorescence intensity was found to be unstable. The elution times required for the decrease in the fluorescence intensity due to the release of the DTAF-monophosphorylated β-casein from the Mn^2+^, Zn^2+^, and Ni^2+^ coordinated Phos-tag acrylamide gel were 12, 120, and 72 s, respectively. We chose Mn^2+^ as the coordinating ion because early elimination would sharpen the peaks in the separation step. Under the optimized conditions described above, we achieved appreciable resolution of the phosphorylated compounds from the Mn^2+^-Phos-tag and the phosphorylated compounds were detected with sharp peaks.

### 2.5. Optimization of Specific Entrapment, Derivatization, and Electrophoretic Separation of Phosphorylated Compounds Labeled with DTAF

We next optimized the online precondition methods utilizing DTAF. In our previous study, we prepared Phos-tag acrylamide gels at the channel crossing of PMMA microchips. However, in this experiment utilizing the PDMS/glass microchip, Phos-tag acrylamide gel was prepared to cover the channel crossing to perform the online fluorescent labeling reaction. As liquids follow the route of least resistance, it is plausible that the incoming biomolecules do not interact with the bulk of the Phos-tag acrylamide gel, especially considering the nm level pore sizes [34]. Experiments were performed to determine whether the sample was efficiently concentrated on the acrylamide gel prepared around the intersection by utilizing our online concentration conditions. Appendix A shows the time-coursed changes in the fluorescence intensity of 10^−9^ M DTAF-monophosphorylated β-casein at various points in the Phos-tag acrylamide gel. The point A of Appendix A illustrates the fluorescence intensity of the non-gel part at the interface with the gel, and the fluorescence intensity of the gel at the other points is indicated by circles. Appendix A show the fluorescence intensity of the gels as measured using a second-harmonic generation (SHG) laser beam (diameter ~30 µm). The fluorescence intensities increased linearly and plateaued eventually. These results are in agreement with those shown in Figure 2. In contrast, the fluorescence intensities did not increase and remained almost the same as before the voltage was applied (Appendix A). These results indicated that the sample solution reached the gel layer in a laminar flow and was concentrated. Figure 5 depicts the time course of the preconcentration of 10^−^^8^ M DTAF-monophosphorylated β-casein with 25 mM sodium acetate buffer (pH 8.5) as the background electrolyte. The concentration had already commenced when a voltage of 150 V was applied through the Phos-tag acrylamide gel after 30 s. The fluorescence intensity increased with the concentration of DTAF-monophosphorylated β-casein over 0.5–3 min and then reached a plateau. This result is in agreement with online concentration of DTAF-labeled monophosphorylated β-casein in Figure 2. These results indicated that the phosphorylated compounds were retained in the Phos-tag gel for 5 min, which allowed online labeling and separation detection following concentration.

### 2.6. Analysis of Phosphorylated Peptides by Online Preconcentration, Derivatization, and Separation

As the final experiment in this study, we investigated a method to complete the pretreatment operations required for the highly sensitive detection of a series of phosphorylated peptides on a PDMS/glass microchip in a few min. After concentrating the unlabeled 10^−9^ M monophosphorylated β-casein solution in the Phos-tag acrylamide gel for 3 min, 10^−7^ M DTAF was introduced into the gel for 5 min for online labeling. Thereafter, the voltage was switched to detect the DTAF-labeled monophosphorylated β-casein that was concentrated and labeled online using the elution buffer. Figure 6 (plot a) portrays the separation of DTAF-labeled monophosphorylated β-casein obtained by the above operation. We detected small and large peaks at 0.8 and 1.1 min, respectively. Figure 6 (plot (b)) portrays the separation of DTAF-labeled dephosphorylated β-casein by alkaline phosphatase. The small peaks detected at ~0.85 min in both the plots (a) and (b) of Figure 6 have approximately the same migration time and peak area and are derived from the excess DTAF that could not be completely removed by the washing operation during the online labeling. On the other hand, the large peak in plot (a) was ascribed to the DTAF-labeled monophosphorylated β-casein because this peak was not detected in plot (b) which was obtained after dephosphorylation by alkaline phosphatase. This result shows that our optimized approach can serve as an efficient method for the derivatization and concentration of phosphorylated compounds.

## 3. Conclusions

We report an improved online preconcentration, derivatization, and separation method for phosphorylated compounds based on the affinity of Phos-tag acrylamide gel in the intersection of a PDMS/glass microfluidic chip toward these compounds. The Phos-tag acrylamide gel was fabricated by irradiation with an LED laser and the photopolymerization was completed within 5 min. The fabricated gel efficiently trapped the phosphorylated compound but not the fluorescent label, DTAF. Thus, the online derivatization and preconcentration of the DTAF-monophosphorylated peptides was mediated by the affinity of the gel to the phosphorylated samples. The total analysis time was approximately 15 min. However, the reported method was applied only to β-casein monophosphorylated peptides. In the near future, we would like to apply it to profile phosphorylated peptides in complex peptide mixtures while improving the peak shape and reproducibility.

## 4. Materials and Methods

### 4.1. Reagents and Materials

Phos-tag acrylamide, acrylamide, *N,N**′*-methylene-*bis*-acrylamide (TEMED), hydroxypropyl cellulose (HPC), *N,N,N**′,N′*-tetramethylethylenediamine, EDTA, and 2,2′-azobis [2-methyl-*N*-(2-hydroxyethyl) propionamide] were obtained from FUJIFIRM Wako Pure Chemical Corporation Co., Ltd. (Tokyo, Japan). FITC was purchased from Tokyo Chemical Industry Co., Ltd. (Tokyo, Japan). β-Casein, monophosphopeptide was obtained from Cosmo Bio Co., Ltd. (Tokyo, Japan). DTAF and NBD-F were purchased from Sigma-Aldrich (Tokyo, Japan). The PDMS base and curing reagent (Sylgard 184) were purchased from Dow Corning Toray Co. Ltd. (Tokyo, Japan), and silicon wafers were procured from Kyushu Semiconductor KAW Co., Ltd. (Fukuoka, Japan). HPC (0.5% *w*/*v*) was added to all the buffers to prevent adsorption of the samples or/and fluorescent reagents to the PDMS/glass chip.

A PDMS microchannel was fabricated by the conventional soft lithography technique. The cross-channel chip consisted of three 5.8 mm and one 27.2 mm-long channels (100 µm width × 30 µm depth), as shown in Figure 1a. After creating the reservoir wells (2.5 mm diameter) with a piercer, these microchannels were directly bonded to the slide-glass via plasma treatment (YHS-R, SAKIGAKE-Semiconductor Co. Kyoto, Japan). An image of the produced PDMS/glass microchip is shown in Figure 1b.

### 4.2. Pre-Column Derivatization of Phosphorylated Peptides with FITC

The monophosphorylated β-casein was derivatized with FITC according to a previously method [35] with slight modifications. A solution of 5 μL of 20 mM FITC in acetone was mixed with 1 mM β-casein phosphorylated peptides in 90 μL of 10 mM sodium phosphate (pH 9.4). The mixture was incubated at 50 °C for 60 min. The reaction mixture was applied to a Toyopearl HW-40F column (1 cm i.d., 10 cm), pre-equilibrated and eluted with water. The effluent was monitored fluorometrically (the excitation and emission wavelengths were 488 and 520 nm, respectively). The FITC-labeled phosphorylated peptides were evaporated to dryness with a centrifugal evaporator (Tomy Seiko Co., Ltd. Tokyo, Japan).

### 4.3. Pre-Column Derivatization of Phosphorylated Peptides with NBD-F

The monophosphorylated β-casein was derivatized with NBD-F according to a previously method [36] with slight modifications. A freshly prepared solution of 0.3 M NBD-F in acetonitrile (5 μL) was added to a solution of 1 mM β-casein phosphorylated peptides in 90 μL of 20 mM phosphate buffer (pH 9.4). The solution was then heated at 40 °C for 30 min. Subsequently, water (100 μL) and dichloromethane (200 µL) were added to the mixture. The solution was shaken vigorously, and the dichloromethane layer was carefully removed. This dichloromethane extraction was repeated twice. Finally, the aqueous layer containing the NBD-labeled phosphorylated peptides was evaporated to dryness with a centrifugal evaporator.

### 4.4. Pre-Column Derivatization of Phosphorylated Peptides with DTAF

The monophosphorylated β-casein was derivatized with DTAF according to a previously reported [37] with slight modifications. A solution of β-casein phosphorylated peptide (1 mM) in 90 μL of 10 mM sodium borate buffer (pH 8.0) was mixed with 5 μL of 20 mM DTAF prepared using a mixture of ethanol and dichloromethane (9:1 *v*/*v*). The solution was heated at 40 °C for 30 min. The reaction mixture was applied to a Toyopearl HW-40F column (1 cm i.d., 10 cm), pre-equilibrated and eluted with water. The eluate was monitored fluorometrically (the excitation and emission wavelengths were 492 and 513 nm, respectively). The DTAF-labeled phosphorylated peptides were evaporated to dryness with a centrifugal evaporator.

### 4.5. ME Apparatus

The ME system used in this study consisted of an inverted epifluorescence microscope (BX 50WI; Olympus Corp., Tokyo, Japan), a 488 nm SHG laser (HPU50222; Furukawa electric CO., Ltd. Tokyo, Japan), a photomultiplier tube (PMT) (H5784MOD; Hamamatsu Photonics, Hamamatsu, Japan), and an HVS448 1500 V (LabSmith, CA, USA) voltage source. We used high numerical aperture (NA) objectives for data collection (20× with NA of 0.7, LCPlanFI; Olympus Corp.), image capture, and irradiation of the laser beam. The laser beam from the SHG laser was focused onto an iris diaphragm and recollimated using sets of convex lenses and block filters (the excitation and emission wavelengths were 450–480 and >515 nm, respectively; U-MWB, Olympus Corp.) to remove unwanted multiple-order energy peaks and ensure transmission of only the central maximum of the laser beam’s diffraction pattern. In our system, the diameters of the laser beams corresponding to magnification values of 20 was 20 μm. The fluorescence emission from the sample was collected directly into a PMT. Fluorescence images of the acrylamide gel were obtained by utilizing a digital microscope (All-in-One fluorescence microscope BZ-X700; Keyence Corp., Osaka, Japan). Photographs of acrylamide gels were obtained using a cooled color charged coupled device (VB-7010; Keyence Corp., Osaka, Japan). During the photopolymerization of the Phos-tag acrylamide, the microchip was placed on the stage of the fluorescence microscope and a 365 nm LED laser (LLS-365, λmax = 365 ± 10 nm, Ocean Photonics, Tokyo, Japan) was focused on the stage for pinpoint irradiation.

### 4.6. Fabrication of a Phos-Tag Preconcentrator Gel on a Cross-Channel Type Microchip

An acrylamide solution (30% T/20% C) consisting of 0.04% Phos-tag acrylamide, 5.8% Tris, 24.0% acrylamide, 6.0% *N,N′*-methylene-bis-acrylamide, 0.75% TEMED, and 0.6% 2,2′-azobis[2-methyl-*N*-(2-hydroxyethyl)propionamide] was de-aerated by connecting to a vacuum line to prevent bubble formation in the fabricated Phos-tag acrylamide gels. The 10 µL of acrylamide solution was poured into the R4 reservoir of the PDMS/glass microchip and introduced to all of the channels by applying pressure and poured acrylamide solution into other reservoirs to prevent the channels from drying out. The channel crossing position, C3 was irradiated with LED for 5 min. Next, all the reservoirs were filled with a 10 mM MnCl_2_ buffer. The coordination of Mn^2+^ to the Phos-tag groups was facilitated by applying a voltage of 0 V to reservoir R8 and 500 V to reservoir R4 for 5 min. The MnCl_2_ solution was replaced with 25 mM sodium acetate buffer (pH 8.5) by pumping it from R4.

### 4.7. Online Affinity ME

Preconcentration of the FITC, DTAF, and NBD-F-phosphorylated compounds on the Phos-tag acrylamide gel in situ fabricated on the PDMS/glass microchip was performed. First, all the reservoirs and channels were filled with 25 mM sodium borate buffer (pH 8.5). The buffer in reservoir R8 was replaced with dilute solutions of the FITC, DTAF, and NBD-F-labeled phosphopeptides. Thereafter, voltages of 0 and 150 V were applied to R8 and R4, respectively. The FITC, DTAF, and NBD-F-labeled labeled phosphopeptides were transferred from R8 to R4 via the three channel crossing points, and the labeled phosphopeptides with an affinity for the Phos-tag acrylamide gel at C3 were entrapped in the acrylamide gel. Components with no affinity, such as excess fluorescence reagents for the Phos-tag, passed through the acrylamide gel.

### 4.8. OnLine Preconcentration, Derivatization, and ME Analysis

Preconcentration of the phosphorylated compounds and their derivatization with DTAF were conducted using the following procedure. A 10^−^^7^ M DTAF solution, 10^−^^8^ M β-casein phosphorylated peptide solution, 100 mM Tris phosphate (pH 7.0)/100 mM EDTA, and 200 mM sodium borate (pH 11.0) were added to R7, R8, R1, and R5, respectively. The other reservoir and all the channels were filled with 25 mM sodium borate (pH 8.5). Next, voltages of 0 and 150 V were applied to R8 and R4 to concentrate the phosphorylated compounds, respectively. After 3 min, the voltages were set to 0 and 100 V on R7 and R4, respectively, to derive the DTAF with phosphorylated compounds. After 5 min, the voltages were set to 0 and 150 V on R2 and R6, respectively, for washing the channel and Phos-tag gel. Thereafter, to separate the DTAF-labeled β-casein phosphorylated peptides, potentials of 0 V and 1500 V were applied for 3 min on R1 and R5, respectively.

## Figures and Tables

**Figure 1 gels-07-00268-f001:**
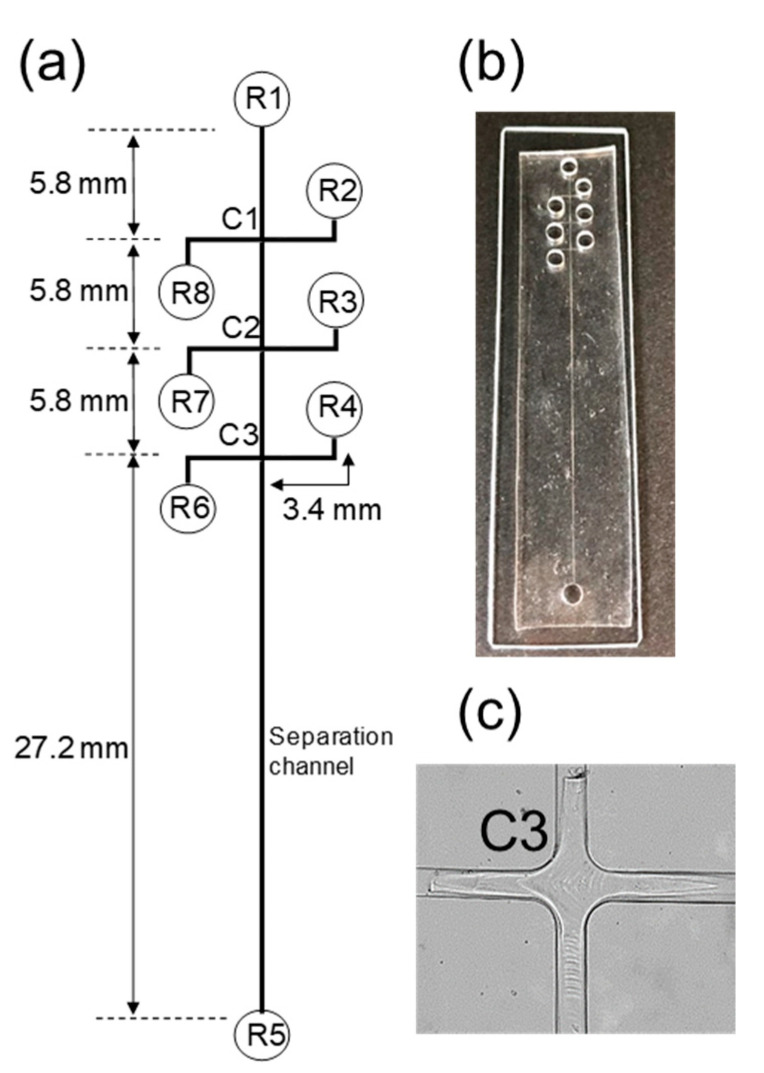
Schematic diagram of polydimethylsiloxane (PDMS)/glass microchip (**a**); photograph of the PDMS/glass microchip (**b**); Phos-tag acrylamide gel formed at the channel crossing point (**c**). R1 = dissociation buffer reservoir; R2 = washing buffer reservoir; R3 = buffer reservoir; R4 = fluorescent reagent waste reservoir; R5 = alkaline buffer reservoir; R6 = fluorescent reagent waste reservoir; R7 = fluorescent reagent reservoir; R8 = sample reservoir, C1–C3 = channel crossing point.

**Figure 2 gels-07-00268-f002:**
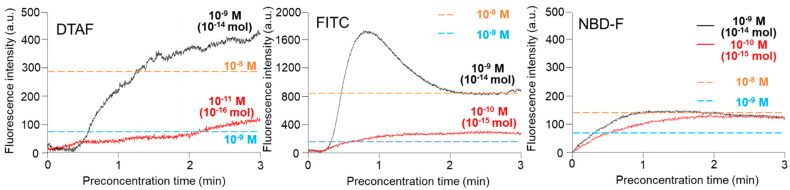
Time-course of fluorescence intensity changes of 5-(4,6-dichlorotriazinyl)aminofluorescein (DTAF), fluorescein isothiocyanate (FITC), and 4-fluoro-7-nitrobenzofurazan (NBD-F)-labeled monophosphorylated β-casein. The concentrations of DTAF, FITC, and NBD-F-labeled monophosphorylated β-casein are indicated on each trace. The orange and blue broken straight lines indicate the fluorescence intensities of DTAF, FITC, and NBD-F-labeled monophosphorylated β-casein at the channel crossing point in the absence of the Phos-tag acrylamide gel.

**Figure 3 gels-07-00268-f003:**
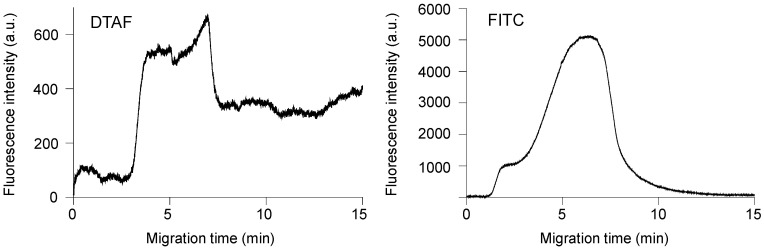
Time-course of fluorescence intensity changes of monophosphorylated β-casein concentrated in Phos-tag acrylamide gel upon introducing 5-(4,6-dichlorotriazinyl)aminofluorescein (DTAF) or fluorescein isothiocyanate (FITC).

**Figure 4 gels-07-00268-f004:**
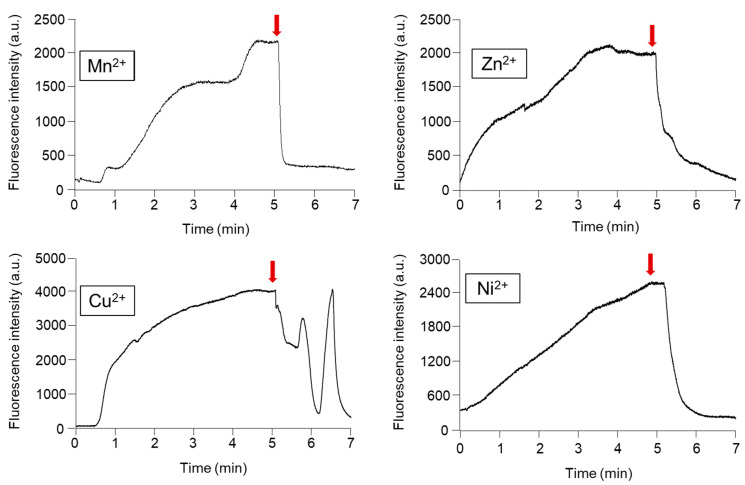
Time course of the changes in the fluorescence intensity due to concentration and release of the 10^−9^ M 5-(4,6-dichlorotriazinyl)aminofluorescein (DTAF)-labeled monophosphorylated β-casein at the channel crossing point in the Phos-tag acrylamide gel utilizing various metal ions. Red arrows indicate the switching voltage for release the preconcentrated DTAF-labeled monophosphorylated β-casein.

**Figure 5 gels-07-00268-f005:**
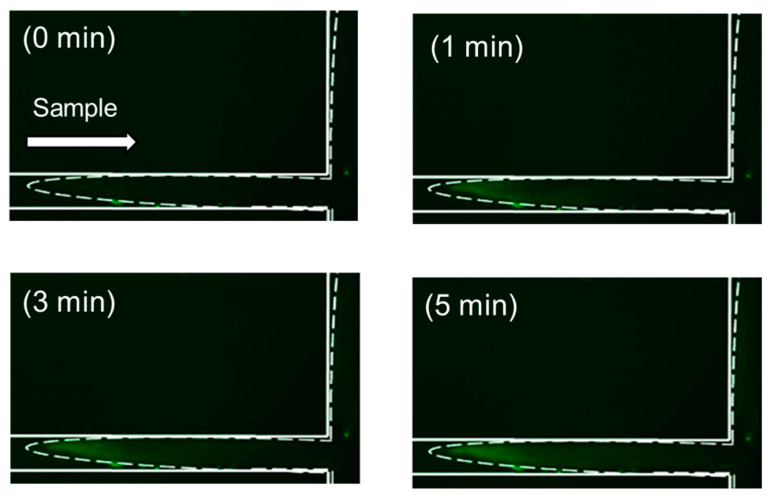
Time-coursed image of fluorescence intensity due to concentration of 10^−^^8^ M 5-(4,6-dichlorotriazinyl)aminofluorescein (DTAF)-labeled monophosphorylated β-casein at the channel crossing in the polydimethylsiloxane/glass microchannels in the presence of in situ fabricated Phos-tag acrylamide gel.

**Figure 6 gels-07-00268-f006:**
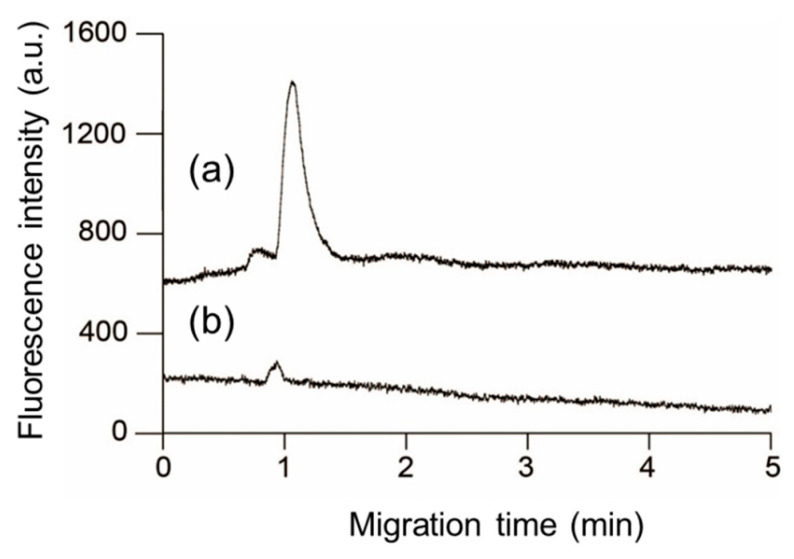
Electrophoretic separation of 5-(4,6-dichlorotriazinyl)aminofluorescein (DTAF)-labeled monophosphorylated β-casein after online preconcentration and derivatization at the channel crossing in the polydimethylsiloxane/glass microchannels with in situ fabricated Phos-tag acrylamide gel (a), and utilizing alkaline phosphatase to dephosphorylate β-casein (b).

## Data Availability

The data presented in this study are available from the corresponding author upon reasonable request.

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
