# Peer review of "In Situ Pinpoint Photopolymerization of Phos-Tag Polyacrylamide Gel in Poly(dimethylsiloxane)/Glass Microchip for Specific Entrapment, Derivatization, and Separation of Phosphorylated Compounds"

_gels, 2021, doi:10.3390/gels7040268_

Round 1

Reviewer 1 Report

I recommend the manuscript “In situ Pin-point Photopolymerization of Phos-tag Polyacrylamide Gel in Poly(dimethylsiloxane)/ Glass Microchip for Specific Entrapment, Derivatization, and Separation of Phosphorylated Compounds” be published after major revisions Gels.  

This submission describes a method for preconcentration, derivatization and separation of phosphorylated compounds. This is done by fabricating a phos-tag acrylamide gel within a junction of a microfluidic devise. Peptides were then concentrated in the gel by applying a voltage, and fluroescin was introduced to derivatize the compounds. This work describes the selection of fluoroescin, selection of key material parameters. Overall the experiments are appropriate for the study however the manuscript is currently lacking significant interpretation / explanations for the data. The authors note difference in fluorescence behavior of their different proposed compounds, however no explanation as to why these differences arise. Similarly very little is presented on the general mechanism by which this device operates. While the authors have published previously on this, there needs to be some background description (it would also help to have a schematic that more clearly describes how this device works).  

Some of the issues associated with the manuscript are described below:

Introduction

  • Line 39: I am confused by the following phrase: “The presence of non-phosphorylated peptides in sample solution suppressed the ionization of phosphorylated peptides, thereby preventing the detection of non-phosphorylated peptides through MS “
    • This is confusing since the authors imply in the previous sentence that the overarching goal is to quantify how much is phosphorylated? (similarly, the sentence after the one listed above also implies such). The authors need to more effectively communicate the challenges with prior art and strategies in this research field.
  • Paragraph starting on line 58 is not actually a paragraph, it should be included in the prior paragraph.
  • The authors make the following statements about their prior work (Line 73): since the gel could be produced only at the channel crossing point, sufficient concentration efficiency could not be achieved.”
    • However, it’s unclear from the introduction section how the present work changes or addresses that issue. The introduction should be significantly revised so the novelty of the present work is more clearly conveyed.

Results and Discussion

  • Figure 1 – The authors should provide more information when introducing this figure, both in the manuscript text and the figure caption. This includes additional description as to what what R1 – R8 correspond to, are employed for, etc.? (this is dicussed briefly later in the text but not in sufficient detail). The reader has information on how the spacing of these different domains was chosen, why are there 8 in total? What was the rational for this design? Considering this platform is the novelty of the work, it needs to be described in appropriate detail.

  • Line 103 – As the authors describe iterating on a final gel formulation, they use the following language:
    • 04% Phos-tag acrylamide in 20%T (percentage of acrylamide compounds in gel 103 solution)/20%C

What do C and T mean in this context? I believe this nomenclature arises from the prior work but it should be briefly summarized here as it is not intuitive.

  • Furthermore, what is meant by “pin-point photopolymerization”? Does this mean on the channel intersection was exposed to light (and thus underwent polymerization?
  • Figure 2 –
    • Why are there no tick-marks on the y-axes? Additionally, can the authors explain why the scaling is so different for DTAF vs. FITC?
    • When discussing these results, the authors need to significantly revise the text to include explanations as to WHY there are different shapes to these fluorescence curves. What is fundamentally different about FITC?
  • Figure 3
    • Why is there such a significant difference in the scales of the two plots?
    • Couldn’t the variations between the behavior of FITC and DTAF be described by the scale variations? When you state that FITC is back at it’s baseline after 13 mins, it’s hard to know if the difference from the baseline is similar to that in fig. 3A.
  • s2 – Why is this figure not included in the manuscript text? Given there is so much discussion on this data it should be in the manuscript? Additionally, why are no y-axes labeled on this figure?
  • Can the authors describe why de-coordination with Cu is unstable? Some rationale for this conclusion should be provided.
  • Fig S3 – Can the authors elaborate why series B and C are different in this Figure? It would be helpful to annotate the image of the intersection to clarify how the data is interpreted.

Author Response

Reviewer 1

I recommend the manuscript “In situ Pin-point Photopolymerization of Phos-tag Polyacrylamide Gel in Poly(dimethylsiloxane)/ Glass Microchip for Specific Entrapment, Derivatization, and Separation of Phosphorylated Compounds” be published after major revisions Gels. 

Reply: We are most grateful to you for the helpful comments on the previous version of our manuscript. We have taken all these comments into account and submit a revised version of our paper.

We hope that our explanations and revisions are satisfactory.

This submission describes a method for preconcentration, derivatization and separation of phosphorylated compounds. This is done by fabricating a phos-tag acrylamide gel within a junction of a microfluidic devise. Peptides were then concentrated in the gel by applying a voltage, and fluroescin was introduced to derivatize the compounds. This work describes the selection of fluoroescin, selection of key material parameters. Overall the experiments are appropriate for the study however the manuscript is currently lacking significant interpretation / explanations for the data. The authors note difference in fluorescence behavior of their different proposed compounds, however no explanation as to why these differences arise. Similarly very little is presented on the general mechanism by which this device operates. While the authors have published previously on this, there needs to be some background description (it would also help to have a schematic that more clearly describes how this device works). 

Reply: Thank you for your advice regarding our paper. Here is the best answer we have prepared. This report mainly achieved the on-line derivatization and preconcentration using DTAF, but we think that it is still necessary to improve the reproducibility.It is necessary to select the material of the microchip itself, so we would like to mention it in the next report.

Some of the issues associated with the manuscript are described below:

Introduction

Line 39: I am confused by the following phrase: “The presence of non-phosphorylated peptides in sample solution suppressed the ionization of phosphorylated peptides, thereby preventing the detection of non-phosphorylated peptides through MS “

This is confusing since the authors imply in the previous sentence that the overarching goal is to quantify how much is phosphorylated? (similarly, the sentence after the one listed above also implies such). The authors need to more effectively communicate the challenges with prior art and strategies in this research field.

Reply:  We are very sorry for this mistake. Although we intended to detect phosphorylated compounds, we have described as non-phosphorylated here. According to this comment we corrected the sentences at Line 40-41 and Line 43 as followed “thereby preventing the detection of non-phosphorylated peptides through MS” to “thereby preventing the detection of phosphorylated peptides through MS”.

“Peptides are commonly derivatized with fluorescent reagents” to “Phosphorylated peptides are commonly derivatized with fluorescent reagents”.

Paragraph starting on line 58 is not actually a paragraph, it should be included in the prior paragraph.

Reply: Thank you for your comment. According to this comment we deleted the paragraph in this part.

The authors make the following statements about their prior work (Line 73): “since the gel could be produced only at the channel crossing point, sufficient concentration efficiency could not be achieved.”

However, it’s unclear from the introduction section how the present work changes or addresses that issue. The introduction should be significantly revised so the novelty of the present work is more clearly conveyed.

Reply: Thank you for your comment. According to this comment, we revised the sentence at Line 69 as followed

“a cross-pattern microchip” to “a single cross-pattern microchip”

In addition, we added the sentences at Line 76 as followed:

Our PDMS/glass microchip has 3 channel crossing points and 8 reservers. We placed five types of solutions such as samples and fluorescent reagents in these eight reservoirs, and achieved concentration, fluorescent labeling, separation, and detection by switching the voltage. The Phos-tag acrylamide gel was formed around one of the three channel crossing point by irradiation with a 365 nm LED laser. By fabricating phos-tag acrylamide gel around the channel crossing point, the volume of the phosphorylated peptide that can be captured was increased by 10 times.

Results and Discussion

Figure 1 – The authors should provide more information when introducing this figure, both in the manuscript text and the figure caption. This includes additional description as to what what R1 – R8 correspond to, are employed for, etc.? (this is dicussed briefly later in the text but not in sufficient detail). The reader has information on how the spacing of these different domains was chosen, why are there 8 in total? What was the rational for this design? Considering this platform is the novelty of the work, it needs to be described in appropriate detail. Line 121

Reply: Thank you for your comment. According this comment, we added the sentence at figure 1 caption as followed;

R1 = dissociation buffer reserver, R2 = washing buffer reserver, R3 = buffer reserver, R4 = fluorescent reagent waste reserver, R5 = alkaline buffer reserver, R6 = fluorescent reagent waste reserver, R7 = fluorescent reagent reserver, R8 = sample reserver

In addition, we added the sentence at Line 99 as followed;

In order to achieve these 4 steps, it is necessary to introduce various solutions into the Phos-tag gel. Also, in step (3), excess fluorescent reagent must be removed from the channel and gel after on-line derivatization for high sensitivity detection. Therefore we fabricated a PDMS microchip with 3 channel cross point and 8 reservoirs, and investigated above 4 steps to achieve only by applying voltage.

Line 103 – As the authors describe iterating on a final gel formulation, they use the following language:

04% Phos-tag acrylamide in 20%T (percentage of acrylamide compounds in gel 103 solution)/20%C

What do C and T mean in this context? I believe this nomenclature arises from the prior work but it should be briefly summarized here as it is not intuitive.

Reply: Thank you for your comment. T and C represent the percentage of acrylamide compounds in gel solution and the percentage of N,N’-methylene-bis-acrylamide in acrylamide compounds, respectively. These explanations were shown in parentheses after T and C, but they are difficult to understand, so we revised this sentence at Line 103 as follows;

“0.04% Phos-tag acrylamide in 20%T (percentage of acrylamide compounds in gel solution)/20%C (percentage of N,N’-methylene-bis-acrylamide in acrylamide compounds)was used for the fabrication of the Phos-tag acrylamide gels in polymethylmethacrylate (PMMA) chips.” to “0.04% Phos-tag acrylamide in 20%T /20%C was used for the fabrication of the Phos-tag acrylamide gels in polymethylmethacrylate (PMMA) chips. T% and C% represent the percentage of acrylamide compounds in gel solution and the percentage of N,N’-methylene-bis-acrylamide in acrylamide compounds, respectively.”

Furthermore, what is meant by “pin-point photopolymerization”? Does this mean on the channel intersection was exposed to light (and thus underwent polymerization?

Reply: Thank you for your comment. As you would expect, pinpoint photopolymerization shows that only the LED irradiation area is changed to gel. Since it was often mistaken for SDS-PAGE due to the keywords of acrylamide gel and electrophoresis, we use pinpoint photopolymerization.

Figure 2 –

Why are there no tick-marks on the y-axes? Additionally, can the authors explain why the scaling is so different for DTAF vs. FITC?

When discussing these results, the authors need to significantly revise the text to include explanations as to WHY there are different shapes to these fluorescence curves. What is fundamentally different about FITC?

Reply: Thank you for your comment. First, we modified Fig.2 and added tick marks the y-axis. We believe that the difference in fluorescence intensity is probably due to our detection system.

Since the SHG laser and filter used this time are compatible with the excitation wavelength and fluorescence wavelength of FITC, it seems that only FITC was detected with high sensitivity.

To clearly describe your point in the text, we added the sentence at Line 183 as followed;

“In Fig. 2, the fluorescence intensity was measured using three fluorescent reagents with the same concentration, but the result was that only FITC had twice the fluorescence intensity. Since we attached a filter for detecting FITC to the microscope, it is considered that the fluorescence intensity of DTAF and NBD-F, which deviate from the excitation wavelength and fluorescence wavelength of FITC, had relatively low.”

Figure 3

Why is there such a significant difference in the scales of the two plots?

Couldn’t the variations between the behavior of FITC and DTAF be described by the scale variations?

Reply: Thank you for your comment. We do not know the reason for a significant difference in the scales of the two plots in detail, but we thought that it is due to the filter explained in Fig. 2 and the feature that FITC is more easily non-specific adsorbed on the Phos-tag gel than DTAF. Also, we tried online labeling and separation many times using FITC, but no peak was detected. We would like to avoid the explanation in the text because we added the explanation of the difference in the fluorescence intensity scale in FIg. 2 and emphasize that the experiment in this section achieved the online labeling in DTAF.

When you state that FITC is back at it’s baseline after 13 mins, it’s hard to know if the difference from the baseline is similar to that in fig. 3A

Reply: Thank you for your comment. According to this comment, we added the Fig. S2 for showing the time course of fluorescence intensity from 1 minute before the start of measurement and 1 minute before the end of measurement. Also, we added the sentence at Line 221 as followed;

In order to show this decrease in fluorescence intensity in detail, Fig. S2 shows the time course of fluorescence intensity from 1 minute before the start of measurement and 1 minute before the end of measurement.

s2 – Why is this figure not included in the manuscript text? Given there is so much discussion on this data it should be in the manuscript? Additionally, why are no y-axes labeled on this figure?

Can the authors describe why de-coordination with Cu is unstable? Some rationale for this conclusion should be provided.

Reply: Thank you for your comment. According to this comment, we have moved FIg. S2 into the text and made it Fig. 4.

We don't know why Cu decoordination is unstable. Since the fluorescence intensity was the highest in this experiment, the ability to capture phosphorylated compounds is the highest. In the next study, we would like to optimize the buffer solution for elution.

Fig S3 – Can the authors elaborate why series B and C are different in this Figure? It would be helpful to annotate the image of the intersection to clarify how the data is interpreted.

Reply: Thank you for your comment. First, according to the reviewer's suggestion, we added the measurement results at the center of the microchip channel crossing point as F. After 3 minutes of introduction, the fluorescence intensity at the F hardly increased. This result is the same in Fig. 5. As the reviewers pointed out, the fluorescence intensity at point B should increase before the fluorescence intensity at point C increases, given the time it takes for the sample to reach the gel. This is due to the variation in each measurement, and the result is that the fluorescence intensity increases at C after B. The results with the highest fluorescence intensity are used for each figure. In fact, we have not obtained satisfactory results regarding the reproducibility of peak area and peak shape in this experiment, so we would like to analyze it in detail in the next experiment.

To clearly this situation, we revised the sentence at Line 330 as followed;

“This method may be utilized to screen and profile phosphorylated peptides in complex peptide mixtures.” to “However, the reported method was applied only to β-casein monophosphorylated peptides. In the near future, we would like to apply it to profile phosphorylated peptides in complex peptide mixtures while improving the peak shape and reproducibility.”

Reviewer 2 Report

This manuscript describes an improved method for online preconcentration, derivatization and separation of phosphorylated compounds based on the affinity of the Phos-labeled acrylamide gel formed at the intersection of a polydimethylsiloxane (PDMS)/glass multichannel microfluidic chip versus some polypeptide compounds.

“However, in the expression “the gels fabricated in the PDMS/glass microchip did not have sufficient mechanical strength(Line 107)” it is written that the gels produced do not have sufficient mechanical strength. It would be appropriate to add a detailed text about how the mechanical strength is determined, how the mechanical strength is studied, and whether it has numerical values.

Author Response

Reviewer 2

This manuscript describes an improved method for online preconcentration, derivatization and separation of phosphorylated compounds based on the affinity of the Phos-labeled acrylamide gel formed at the intersection of a polydimethylsiloxane (PDMS)/glass multichannel microfluidic chip versus some polypeptide compounds.

Reply: We are most grateful to you for the helpful comments on the previous version of our manuscript. We have taken the comment into account and submit a revised version of our paper.

We hope that our explanations and revisions are satisfactory.

“However, in the expression “the gels fabricated in the PDMS/glass microchip did not have sufficient mechanical strength(Line 107)” it is written that the gels produced do not have sufficient mechanical strength. It would be appropriate to add a detailed text about how the mechanical strength is determined, how the mechanical strength is studied, and whether it has numerical values.

Reply: We are very sorry for the misleading expression. Here, we are experimenting with whether breakage occurs by applying a voltage of 1 kV, and we have not investigated the mechanical strength.

To clearly this situation, we deleted the “mechanical” in this text, and revised the sentence at Line 107 as followed;

“However, the gels fabricated in the PDMS/glass microchip did not have sufficient mechanical strength.” to “However, the gels fabricated in the PDMS/glass microchip broke when a voltage of 1 kV was applied.”

Round 2

Reviewer 1 Report

I recommend the manuscript “In situ Pin-point Photopolymerization of Phos-tag Polyacrylamide Gel in Poly(dimethylsiloxane)/ Glass Microchip for Specific Entrapment, Derivatization, and Separation of Phosphorylated Compounds” be published after minor revisions to Gels.  

This submission describes a method for preconcentration, derivatization and separation of phosphorylated compounds. This is done by fabricating a phos-tag acrylamide gel within a junction of a microfluidic devise. Peptides were then concentrated in the gel by applying a voltage, and fluroescin was introduced to derivatize the compounds. Overall, the design of the device and the experiments are appropriate for the study. Furthermore, the authors have improved their discussion so some of the results and interpretation are better understood. However, there are still aspects of the study that are not fully described, and the manuscript does have a significant number of grammatical errors. These should be addressed prior to publication.

Some of the issues associated with the manuscript are described below:

Grammar:

  • There are numerous errors (lines 38, 54, 58, 59, 61, 79, 108-112, 116, 131, 142, 145, 191..). Most of these deal with verb/subject agreement, use of the word reserver instead of reservoir. Etc.

Introduction

  • The authors make the following statements about their prior work (Line 73): since the gel could be produced only at the channel crossing point, sufficient concentration efficiency could not be achieved.”
    • However, it’s unclear from the introduction section how the present work changes or addresses that issue and despite revisions it is still unclear to me.
    • I know they now say “since it is around the channel crossing point the amount captured increased 10 fold….” But it still sounds like it’s done the exact same way as before.
    • The authors say they addressed this issue by saying a “cross-pattern microchip” as opposed to a “single cross-pattern microchip”. However, the way the manuscript is written it implies that have the gel only at the cross-point is a limiting factor. Considering the gel is formed at a junction in this study, it’s not clear that this is a major improvement.

Results and Discussion

  • What is the difference between the upper and the lower panel of Figure 2? Is one being deleted compared to the other?
  • 3 – The authors need to provide some comment or discussion on the difference in the scale of the two sets of data presented. Is this caused by the filter mentioned in the discussion of Figure 2? Overall, the justification for employing DTAF is weak given the data is much noisier and has a lower intensity.
  • 4 – can you comment on the lack of stability for Cu2+? Why does this happen? Some discussion should be included.
  • 5 – is there really a difference in intensity between 1 / 3 / and 5 mins? It’s hard to see in the images and quantitative data should be provided to support this claim.
  • Line 307 – fig. 5 should be Fig 6

Author Response

Reviewer 1

I recommend the manuscript “In situ Pin-point Photopolymerization of Phos-tag Polyacrylamide Gel in Poly(dimethylsiloxane)/ Glass Microchip for Specific Entrapment, Derivatization, and Separation of Phosphorylated Compounds” be published after minor revisions to Gels. 

This submission describes a method for preconcentration, derivatization and separation of phosphorylated compounds. This is done by fabricating a phos-tag acrylamide gel within a junction of a microfluidic devise. Peptides were then concentrated in the gel by applying a voltage, and fluroescin was introduced to derivatize the compounds. Overall, the design of the device and the experiments are appropriate for the study. Furthermore, the authors have improved their discussion so some of the results and interpretation are better understood. However, there are still aspects of the study that are not fully described, and the manuscript does have a significant number of grammatical errors. These should be addressed prior to publication.

Reply: We are most grateful to you for the helpful comments on the previous version of our manuscript. We have taken all these comments into account and submit a revised version of our paper.

We hope that our explanations and revisions are satisfactory

Some of the issues associated with the manuscript are described below:

Grammar:

There are numerous errors (lines 38, 54, 58, 59, 61, 79, 108-112, 116, 131, 142, 145, 191..). Most of these deal with verb/subject agreement, use of the word reserver instead of reservoir. Etc.

Introduction

Reply:  We are very sorry for these mistakes. We corrected all grammatical errors. Please see the revised text.

The authors make the following statements about their prior work (Line 73): “since the gel could be produced only at the channel crossing point, sufficient concentration efficiency could not be achieved.”

However, it’s unclear from the introduction section how the present work changes or addresses that issue and despite revisions it is still unclear to me.

I know they now say “since it is around the channel crossing point the amount captured increased 10 fold….” But it still sounds like it’s done the exact same way as before.

The authors say they addressed this issue by saying a “cross-pattern microchip” as opposed to a “single cross-pattern microchip”. However, the way the manuscript is written it implies that have the gel only at the cross-point is a limiting factor. Considering the gel is formed at a junction in this study, it’s not clear that this is a major improvement.

Reply: Thank you for your comment. As the reviewer 1 point out, the current Introduction focused on online concentration, and this online concentration method is based on the same principles as previous studies and has not changed significantly. Since we mainly described online fluorescent labeling method in this manuscript, we have revised the manuscript significantly as shown below.

Line 71; In addition, since the gel could be produced only at the channel crossing point, sufficient concentration efficiency could not be achieved. → deleted.

Line 73; “Here, we describe an improved method for the online preconcentration, derivatization,” to “Here, we describe an improved method for the preconcentration, online derivatization,”

Line 83; “increased by 10 times.” to “increased by 10 times and utilized as a wide reaction field for online fluorescent derivatization.”

Line 83 and 84; phosphorylated peptides → unlabeled phosphorylated peptides

Results and Discussion

What is the difference between the upper and the lower panel of Figure 2? Is one being deleted compared to the other?

Reply: Thank you for your comment. The upper figure is the deleted figure during Major Revision. The previous figure remained because it was corrected by Track Changes mode. It has disappeared in this revision.

3 – The authors need to provide some comment or discussion on the difference in the scale of the two sets of data presented. Is this caused by the filter mentioned in the discussion of Figure 2? Overall, the justification for employing DTAF is weak given the data is much noisier and has a lower intensity.

Reply: Thank you for your comment. First, the difference in the scale of the two sets of data is due to the difference in filters as pointed out by Reviewer 1. To clearly this situation, we added the new reference and sentences at Line 223 as followed;

There was a large difference in the fluorescence intensity obtained in Fig. 3, which is probably due to the filter set explained in Fig. 2.

As you pointed out, if we replace the filter set, we may be able to select the most suitable fluorescent reagent than DTAF. In this experiment, FITC and NBD-F also tried to detect unlabeled samples by the same operation as DTAF, but no peak was obtained. Here, we reported that unlabeled phosphorylated peptides by online labeling using DTAF. However, we think the sensitivity, reproducibility, and peak shape need to be improved, so we would like to achieve these in the next experiment.

4 – can you comment on the lack of stability for Cu2+? Why does this happen? Some discussion should be included.

Reply: Thank you for your comment. We re-examined the reason for the lack of stability for Cu2+ and found that the ability of Phos-tag to capture phosphorylated compounds differs depending on the phosphorylation site and molecular weight. We think this is because β-casein has a higher capture capacity than other metals by using copper.

To clearly this situation, we added the new reference and sentences at Line 249 as followed;

The ability of Phos-tag to capture phosphorylated compounds differs depending on the phosphorylation site and molecular weight [33]. When ~

[33] Asakawa, D.; Miyazato, A.; Rosu, F.; Gabelica V. Influence of the metals and ligands in dinuclear complexes on phosphopeptide sequencing by electron-transfer dissociation tandem mass spectrometry. Phys. Chem. Chem. Phys., 2018, 20, 26597-26607, doi: 10.1039/C8CP04516J

5 – is there really a difference in intensity between 1 / 3 / and 5 mins? It’s hard to see in the images and quantitative data should be provided to support this claim.

Reply: Thank you for your comment. The below figure (please see the attachment file) shows the time-course of fluorescence intensity changes of DTAF-labeled monophosphorylated β-casein under these conditions. The figure shows the same behavior as the one obtained in Fig. 2. As you pointed out, the difference between 1 minute and 3 minutes is particularly difficult to understand, and the fluorescence intensity increased from around 30 seconds when the voltage was applied.

To clearly this situation, we revised and added the sentences at Line 288 and 290, respectively as followed;

Line 288  “The concentration had already commenced when a voltage of 150 V was applied through the Phos-tag acrylamide gel after 1 min. The fluorescence intensity increased with the concentration of DTAF-monophosphorylated β-casein over 1–3 min and then reached a plateau.” to “The concentration had already commenced when a voltage of 150 V was applied through the Phos-tag acrylamide gel after 30 sec. The fluorescence intensity increased with the concentration of DTAF-monophosphorylated β-casein over 0.5–3 min and then reached a plateau.”

Line 290  This result is in agreement with on-line concentration of DTA-labeled monophosphorylated β-casein in Fig. 2.

Line 307 – fig. 5 should be Fig 6

Reply:  We are very sorry for this mistake. We changed it.
